# Validation of Pressure-Sensing Insoles in Patients with Parkinson’s Disease during Overground Walking in Single and Cognitive Dual-Task Conditions

**DOI:** 10.3390/s22176392

**Published:** 2022-08-25

**Authors:** Monica Parati, Matteo Gallotta, Manuel Muletti, Annalisa Pirola, Alice Bellafà, Beatrice De Maria, Simona Ferrante

**Affiliations:** 1Neuroengineering and Medical Robotics Laboratory, Department of Electronics, Information and Bioengineering, Politecnico di Milano, 20133 Milan, Italy; 2Istituti Clinici Scientifici Maugeri IRCCS, 20138 Milan, Italy

**Keywords:** Parkinson’s disease, gait analysis, insoles, wearable sensors, dual tasking, validation, rehabilitation

## Abstract

There is a need for unobtrusive and valid tools to collect gait parameters in patients with Parkinson’s disease (PD). The novel promising tools are pressure-sensing insoles connected to a smartphone app; however, few studies investigated their measurement properties during simple or challenging conditions in PD patients. This study aimed to examine the validity and reliability of gait parameters computed by pressure-sensing insoles (FeetMe^®^ insoles, Paris, France). Twenty-five PD patients (21 males, mean age: 69 (7) years) completed two walking assessment sessions. In each session, participants walked on an electronic pressure-sensitive walkway (GaitRite^®^, CIR System Inc., Franklin, NJ, USA) without other additional instructions (i.e., single-task condition) and while performing a concurrent cognitive task (i.e., dual-task condition). Spatiotemporal gait parameters were measured simultaneously using the pressure-sensing insoles and the electronic walkway. Concurrent validity was assessed by correlation coefficients and Bland–Altman methodology. Test–retest reliability was examined by intraclass correlation coefficients (ICC) and minimal detectable changes (MDC). The validity results showed moderate to excellent correlations and good agreement between the two systems. Concerning test–retest reliability, moderate-to-excellent ICC values and acceptable MDC demonstrated the repeatability of the measured gait parameters. Our findings support the use of these insoles as complementary instruments to conventional tools during single and dual-task conditions.

## 1. Introduction

Walking impairments are common in patients with Parkinson’s disease (PD), and they become more debilitating as the disease progresses [1]. Bradykinesia, rigidity, shuffling, loss of automaticity, freezing, and postural instability are recurrent symptoms that encompass PD walking [2]. They may contribute to shortened stride length, slower walking speed, increased double support time, and stride-to-stride variability, compared with healthy subjects [1]. Even early in the disease course, PD patients have deficits in executive functions and in the ability to divide attention resources [3]. When patients walk while performing a concurrent cognitive task, the requested additional attention load affects the patients’ motor functions, exacerbating gait abnormalities [4]. Some studies demonstrated that patients with compromised dual-task performance during gait had a higher fall risk [4] and reduced functional capacities [5], leading to disabling consequences during daily life activities [4]. For its higher sensitivity on detecting gait abnormalities than single tasking [4,6] and relevant impact on daily life activities, dual-task walking is gaining ground as a physical therapy intervention, as well as an assessment tool to study motor-cognitive inference in people with PD [4,7].

In addition to clinical scales, several quantitative methods allow to examine PD gait abnormalities in single and dual-task conditions. Optical motion analysis systems and pressure-sensitive walkways are the reference tools to detect gait events with high accuracy [8,9], but their applicability is only possible in specialized settings due to their limited portability, expensive costs, and intensive commitment of resources [8,9]. Moreover, they are applied during sporadic visits and could not continuously give feedback about the patient’s functional performances in a complex context, as a daily life setting. Wearable technology, namely any electronic device worn by a person, has made testing daily life walking more feasible [10,11].

Among this category, different insole prototypes have been proposed. They are instruments used to perform proprioceptive training using biofeedback or to collect balance and gait parameters for monitoring PD patients, or other neurological, orthopedic, diabetic users [12]. In recent years, some novel thin and lightweight solutions were developed to compute gait parameters. Among the lastly developed and commercialized insoles, FeetMe^®^ insoles (FeetMe, Paris, France) present some advantages. They integrate inertial measurement units (IMU) and pressure sensors inside the soles and do not present wires or other external modules, visible over the clothes, as occurs for other instrumented insole systems, such as Medilogic, LoadSole^®^, Tekscan, and Pedar system [13,14]. Furthermore, they do not require a connection box connected to a computer to transmit data and they have a real-time feedback of the gait parameters on the smartphone application, contrary to what occurs for PodoSmart and Moticon OpenGo insoles [15,16].

Indeed, FeetMe^®^ insoles compute spatio-temporal gait parameters through embedded algorithms and collect them through a smartphone application [17]. All these characteristics lead to better ergonomics, portability, and ease of use, that could enhance the end-users’ compliance and acceptance to wear over long-term monitoring [12].

Moreover, previous findings demonstrated the acceptable validity and reliability of the collected gait parameters in middle-aged [17] and community-dwelling older adults [18]. However, limited data were available on pathological conditions. A previous article demonstrated good concurrent validity and reliability in patients with chronic stroke [19]. These previous validation studies did not include the PD population, and they mainly restricted their examination during single-task walking.

We hypothesized that FeetMe^®^ insoles would provide valid gait measurements in PD patients, with respect to a reference pressure-sensitive electronic walkway and highly repeatable measures between two testing sessions. Finally, we hypothesized that they could be an accurate and precise system when tested during a more challenging condition, such as cognitive dual-task walking.

Therefore, the study aims are to determine the concurrent validity and test–retest reliability of FeetMe^®^ insoles for measuring gait spatiotemporal parameters in PD patients, during over-ground straight-line walking in single and cognitive dual-task conditions.

## 2. Materials and Methods

### 2.1. Study Design

This was a cross-sectional study conducted at the IRCCS Istituti Clinici Scientifici Maugeri (Milan, Italy). Procedures were conducted according to the Declaration of Helsinki and approved by the Hospital Research Ethics Committee (2574CE, 1 September 2021).

### 2.2. Participants

A convenience sample of individuals with Parkinson’s disease undergoing outpatient rehabilitation at IRCCS Istituti Clinici Scientifici Maugeri (Milan, Italy) participated in this study. The inclusion criteria were: (i) age higher than 18 years, (ii) diagnosis of idiopathic PD, (iii) with mild-to-moderate disease severity (modified Hoehn and Yahr (mH&Y) scale: 1–3), and (iv) stable physical and pharmacological condition during the study period. The exclusion criteria were (i) insufficient cognitive function to follow given instructions (Mini-Mental State Examination (MMSE) equal or lower than 24), and (ii) any other comorbidities interfering with gait.

### 2.3. Instrumentation

This study adopted six pairs of FeetMe^®^ Insoles (European sizes 37–42, insole weight: 80–120 g, medical device approval: Class Im CE(93/42/EC) and Class I FDA 510(k) exempt) [20]. Each insole (Figure 1a) includes 18 capacitive pressure sensors (sampling frequency: 110 Hz, pressure cell area: 15 mm^2^), a 6-axis IMU (sampling frequency: 146 Hz, accelerometer range: 8 g, gyroscope range: 1000 dps), a rechargeable Li-ion battery (standard charging current: 110 mAh), and a built-in recording data memory [17,20].

### 2.4. Procedure

After participants signed the informed consent, a clinical evaluation and a gait assessment were performed in the dopaminergic medication ON state. The clinical evaluation included MMSE, the motor subscale of Unified Parkinson’s Disease Rating Scale (UPDRS-part III) and the mH&Y scale. Participants are then invited to perform the gait assessment by an assessor (Visit 1). They were requested to walk over the electronic walkway (GAITRite^®^ Walkway System [21]) with insoles inside their shoes and complete six walking trials. For each trial, participants were invited to start walking one meter before the walkway active area and to continue walking one meter past the walkway active area without slowing.

The first three trials consisted of walking bouts at a self-selected comfortable speed without other concurrent demands (i.e., single-task). In the other three walking trials, participants completed a cognitive dual-task assessment. They were asked to walk at their usual pace while serially subtracting the number 3 from a randomly selected number from 50 to 100. No explicit (cognitive or walking) task prioritization instructions were given to the participants.

One minute of rest was administered between tasks and extra time was allowed if the participants asked for it. Familiarization trials for each task condition were also performed.

Participants repeated the walking trials in the same conditions, within one week (Visit 2) (Figure 1c,d). They repeated the trials in the dopaminergic medication ON state at the same hour of the day within one week to minimize possible PD motor fluctuation symptoms. All patients kept the same shoes in both testing sessions. The same assessor examined the patient in both testing sessions. Possible discomforts and adverse events while wearing insoles were recorded at the end of the walking trials.

### 2.5. Data Analysis and Statistical Methods

All data and statistical analyses were performed using MATLAB program (R2021b, MathWorks Inc., Natick, MA, USA) and SPSS software v28 (SPSS Inc., Chicago, IL, USA). The level of significance was set to *p* < 0.05.

Spatiotemporal gait parameters for all the patients’ strides during single-task and dual-task trials were separately extracted from insoles and electronic walkways. Their descriptive statistics are computed and their mean and standard deviation were presented. Gaussian distribution of the data was tested using Shapiro–Wilk test. Student’s paired *t*-test or Wilcoxon signed rank test were applied to test significant differences between task condition (single- vs. dual-task), as appropriate. Relative reliability between the mean values of gait parameters in the two testing sessions was investigated by calculating the intraclass correlation coefficient (ICC) with 95% confidence interval (CI). ICC was calculated based on a mean-rating, absolute-agreement, 2-way mixed-effects model [22]. ICCs could vary between 0 and 1 and were commonly interpreted as “poor” for values below 0.50, moderate between 0.50 and 0.75, good between 0.75 and 0.90, and excellent above 0.90 [22]. Absolute reliability indexes were examined in terms of raw units (MDC) and percentages to the parameter mean value (MDC%) [23]. MDC was calculated at 95% level using the formula:(1)$$\mathrm{MDC}=1.96\times \sqrt{2}\times {\mathrm{SD}}_{\mathrm{Pooled}}\times \sqrt{1-\mathrm{ICC}}$$
where ${\mathrm{SD}}_{\mathrm{Pooled}}$ is the pooled standard deviations of the two testing sessions. MDC values were considered acceptable if lower than the 30% of the parameter mean value (i.e., MDC% < 30%) [9].

Student’s paired *t*-test or Wilcoxon signed rank test were used, as appropriate, to verify the absence of systematic errors between the two sessions (Visit T1 vs. Visit T2) [24].

Two types of analysis were made to investigate the concurrent validity. Firstly, the comparison was made between the mean values of all strides for each participant (hereafter: mean stride data analysis) and secondarily between individual footfall values across the whole cohort (hereafter: stride-to-stride data analysis). For both analyses, Pearson’s correlation coefficient or Spearman’s rank correlation coefficients were calculated to measure the correlation between the walkway- and smartphone-derived measures. The degree of correlation was interpreted as <0.30 being negligible, 0.30–0.50 low, 0.50–0.70 moderate, 0.70–0.90 good, and 0.90–1.00 excellent [25]. Further, Bland and Altman analysis was done to extract the systematic mean bias and the 95% limits of agreement (LoA) between the two systems [26].

The sample size was estimated based on a priori power analysis, that showed 22 was the minimum number of participants to establish that a correlation coefficient of 0.80 was significantly different from a minimally acceptable coefficient of 0.50, considering an α = 0.05 and 1 − β = 0.80 [27]. The sample size was increased to 25 to allow for a possible 10% drop-out rate.

## 3. Results

Twenty-five patients (21 males, mean age 69 (7) years) were included in the study. Their sociodemographic and clinical characteristics were reported in Table 1.

All the participants completed the walking trials and no insoles discomfort or adverse events were reported. The two testing sessions were separated by 3 (2) days.

Figure 2 displays significant differences in gait parameters computed by insoles between the two testing conditions.

### 3.1. Test–Retest Reliability

Table 2 includes the test–retest reliability findings of the gait parameters collected by pressure-sensing insoles. In detail, the table shows the mean of gait parameters in the two walking sessions and the absence of systematic errors between the first and second measurement in both systems, as well as the ICC and MDC findings.

ICC values of the insoles were within the range of 0.88–0.97, and most of them (89%) were higher than 0.90 during single-task walking. Similar findings were found in the dual-task condition (ICC range: 0.80–0.95), in which 89% of the parameters had an ICC higher than 0.90. During the single-task condition, the MDC% values were acceptable and in the range of 7.6–16.0%. In dual-task condition, the MDC% showed suboptimal results for double support time (38.2%) and higher but still acceptable results in the other gait parameters (13.5–25.5%).

In both conditions, the reference systems showed reliability results comparable to insoles findings, as shown in Table 3.

### 3.2. Concurrent Validity

In the mean stride data analysis (Table 4), correlation coefficients ranged from 0.73 to 1.00 in both single and dual-task conditions.

Similar findings were achieved in the stride-to-stride data analysis (Table 5). Indeed, the stride-to-stride analysis revealed correlation coefficients between 0.54 and 0.98.

The levels of agreement (Table 4 and Table 5) of the temporal parameters showed a mean absolute bias lower than 0.05 s. Instead, in the parameter involving spatial information, Bland and Altman analysis revealed that insoles measurements of stride velocity and length were biased towards lower values than walkway measurements. Similar findings were achieved in the dual-task condition, but a slight increase in the width of the limits of agreement was noticed.

## 4. Discussion

The study investigated the concurrent validity and test–retest reliability of gait spatiotemporal parameters computed by FeetMe^®^ plantar pressure-sensing insoles against a reference pressure-sensitive walkway. The study findings revealed that all the enrolled twenty-five patients completed the study protocol and did not report any specific discomfort or adverse events while wearing the insoles inside their shoes. It demonstrated that these insoles were well-tolerated instruments for clinical application. In addition, the study indicated that many spatiotemporal parameters were accurate and repeatable during both single-task and dual-task walking.

As noticed in previous studies [28,29,30], dual-tasking determined a generalized worsening of spatiotemporal gait parameters in patients with PD. The loss of walking automaticity, the deficits in executive function, and the difficulties in dividing attention are common in Parkinson’s disease [3,31]. When patients allocated additional attention demands to walk, gait abnormalities and phenotypic heterogeneity within the PD population are enhanced [32]. As observed in this study, this results in slower gait speed, shorter strides, longer double support time, and increased stride variability during dual-task walking.

With regard to reliability, the gait parameters did not significantly change within the two sessions, revealing the absence of systematic errors. The relative reliability was higher than 0.75, indicating a reliability from moderate to excellent in both task conditions. When discussing ICC findings with respect to previous research articles on the same instrument, several factors could impact the resulting ICC values, including the heterogeneity of the study sample, the applied statistical model, and the measurement protocols [22,33].

In this study, ICCs were generally lower than previous articles focused on FeetMe^®^ insoles [18,19]. Lunardini and colleagues [18] showed values higher than 0.89 in a cohort of older adults without pathologies affecting gait. However, they did not examine the repeatability between a day-to-day measurement. Indeed, their protocol included two consecutive 3 min bouts during single-task walking. Farid and colleagues [19] applied a protocol more similar to the current study. They asked patients with chronic stroke to walk on an electronic walkway and showed excellent reliability results (ICC: 0.94–0.97) [19]. However, a complete and direct comparison of the results was not feasible. Farid and colleagues [19] did not report the temporal parameters (i.e., single-support and double support time), which showed the lowest reliability properties in our study. These parameters required an accurate detection of both foot-strike and toe-off events. Patients with Parkinson’s disease, even in a mild or moderate disease stage, could assume an impaired foot loading pattern as a possible consequence of adaptive mechanisms to avoid unsteadiness during walking [34]. The impaired foot loading pattern might have increased the uncertainties around the gait event estimates during different testing sessions.

For clinical interpretability, we present a set of MDC values for parkinsonian gait. The MDC analysis is one of the most common and useful absolute reliability methods. It helps to know the lowest real within-person change on a gait parameter beyond the measurement error [33,35]. MDC analysis also determines whether an instrument is sensitive enough to detect true changes in the gait parameters [33,35]. In the single-task condition, the MDC values were small and acceptable, whereas in the dual-task condition, the MDC showed suboptimal results for double support time and higher but still acceptable results in the other gait parameters. In both conditions, the reference systems showed MDC% results comparable to insoles results, and in lines with findings of a recent review on pressure-sensitive walkways [9]. Concerning system validity, the previous validation study by Jacobs et al. [17] performed a mean stride data analysis to examine the agreement between FeetMe^®^ insoles and the reference walkway in a group of middle-aged subjects during single-task walking. Their findings demonstrated high correlations (≥0.76) among the two systems [17]. Farid et al. [19] found moderate to very high correlations (≥0.57) testing the validity between insoles- and walkway-derived measures in a group of patients with chronic stroke during the single-task condition. In the present study, good-to-excellent correlation coefficients were obtained in stride velocity, cadence, stride length, and stride time in both task conditions (≥0.87). The insoles provided acceptable estimates of the other temporal parameters (i.e., single limb support, double support, stance time, swing time, and step time) with correlations higher than 0.50. Bland–Altman analysis comparing insoles and walkway-derived measures generally revealed good measurement agreement. Overall, the dual-task condition exhibited slightly wider limits of agreement than the single-task condition, in line with previous studies [36,37]. However, this between-methods comparison yielded similar relative mean differences in both task conditions. The highest levels of agreement were shown by the temporal parameters, which generally have a mean absolute bias lower than 0.05 s. Concerning parameters that involved spatial information, the insoles tended to underestimate the true values of stride length and stride velocity. However, their mean biases were still acceptable because they were low in relation to the gait parameter scale.

Therefore, the proposed insoles proved their ability to measure valid and reliable data in PD patients. In addition, they have several advantages over previous instrumented insoles tested on PD patients [38,39,40]. In more detail, they integrate pressure and IMU sensors in a thin layer and embedded algorithms that computed and wirelessly sent gait parameters in real-time to a smartphone. These elements maximize patients’ comfort and speed up gait assessment during busy everyday clinical practices, ultimately favoring their use in clinics and during long-term home monitoring. In addition, the attractive opportunity to integrate these devices with other wearable biosensors, such as those offered by electrochemical sensing technology [41,42], into a broad IoT network might enhance the possibility of a quantitative, continuous, and synchronized monitoring of PD-related motor symptoms and PD-treating drug levels to manage patient’s motor fluctuations and provide personalized patient-centered care [42]. However, further studies should test the long-term stability and usability of these insoles in remote environments.

The present study has some limitations to be considered. The protocol did not include age-matched healthy controls, precluding possible between-group comparisons. Furthermore, the number of correct responses and errors were not recorded during the cognitive dual-task condition, precluding further analyses on task prioritization strategies in the examined population. The sample size was inadequate to perform additional subgroup analyses to examine the measurement properties at different disease severity stages. Most of the included patients were not severely affected because their mH&Y score was equal or lower than 2. However, they were not free from motor problems since their mean UPDRS motor score was 16.9 (3.7). It is possible that gait deficits in more severely impaired patients could influence the measurement properties. Substantial deviations from the healthy gait template due to abnormal gait waveforms could lead to falsely undetected stride, potentially biasing the results. Future works should focus on these patients, specifically those with the higher risk of falling and freezing of gait but also extend this work to other understudied pathologies. Finally, our reference system allows only short bouts of straight walking inside a controlled clinical environment. Turning or other free-living movements should be considered in future studies to assess the accuracy and precision in less constrained movement situations.

## 5. Conclusions

This study showed the validation of the gait parameters from FeetMe^®^ insoles under single and cognitive dual-task conditions, with promising results for future applications. Their flexibility, portability and ease of use make them relevant tools for large-scale analyses of patients with Parkinson’s disease. They could concurrently be used in clinical settings, not equipped with sophisticated and cumbersome gait analysis equipment, and free-living contexts. However, these findings refer only to over-ground straight-line walking in single and cognitive dual-task conditions recorded on ambulatory PD patients with mild-to-moderate disease severity. Our findings need to be examined in more severely impaired patients and generalized to other challenging and complex testing conditions, such as walking along curvilinear trajectories or in unsupervised settings.

## Figures and Tables

**Figure 1 sensors-22-06392-f001:**
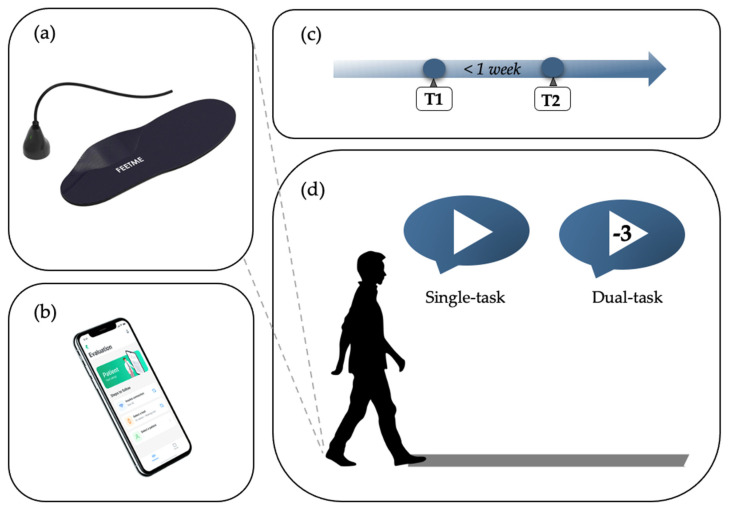
Schematic representation of the instrumentation and experimental procedure applied. (**a**) Pressure−sensing FeetMe^®^ insoles [20]; (**b**) the FeetMe^®^ evaluation mobile app to collect gait data from insoles [20]; (**c**) the gait assessment timeline; (**d**) the experimental protocol applied during a gait assessment.

**Figure 2 sensors-22-06392-f002:**
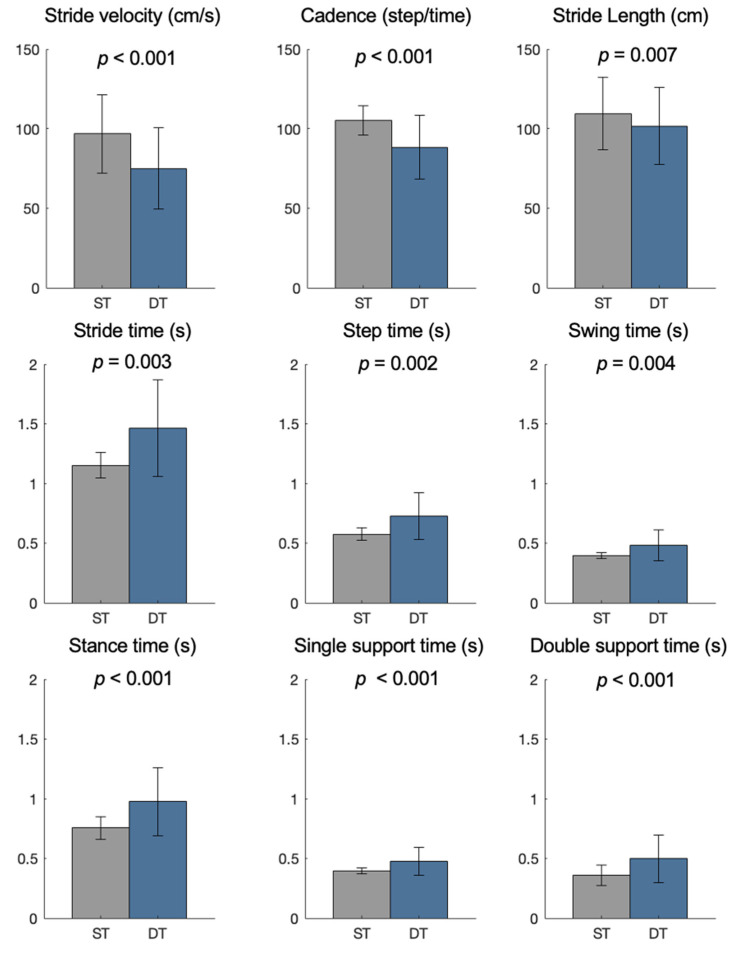
Spatial and temporal gait parameters measured by pressure-sensing insoles in the single-task (ST) and dual-task condition (DT).

**Table 1 sensors-22-06392-t001:** Sociodemographic and clinical characteristics of the participants.

Characteristics	N = 25
Age	69 (7)
Gender (Male/Female)	21/4
Weight (kg)	75.0 (10.2)
Height (cm)	169.2 (8.8)
BMI (kg/m^2^)	26.2 (3.4)
Duration of the disease (years)	4 (3)
UPDRS-part III	16.9 (3.7)
mH&Y score	2.0 (0.6)

Values are reported as mean (standard deviation) or absolute number. BMI: body mass index, UPDRS-part III: motor subscale of the Unified Parkinson’s Disease Rating Scale, mH&Y: modified Hoehn and Yahr.

**Table 2 sensors-22-06392-t002:** Test–retest reliability of spatio-temporal parameters computed by pressure-sensing insoles during single and dual-tasking conditions.

Condition	Gait Parameter	Visit T1 Mean (SD)	Visit T2 Mean (SD)	T1 vs. T2 *p*-Value	ICC [95%CI]	MDC	MDC%
Single-task	Stride velocity (cm/s)	95.6 (24.4)	92.7 (22.7)	0.185	0.95 [0.88–0.98]	15.04	16.0
	Cadence (step/min)	105.4 (9.3)	104.4 (9.9)	0.376	0.91 [0.80–0.96]	7.94	7.6
	Stride length (cm)	107.5 (22.2)	105.6 (21.3)	0.248	0.97 [0.92–0.99]	11.12	10.4
	Step time (s)	0.58 (0.06)	0.59 (0.06)	0.136	0.92 [0.82–0.97]	0.04	7.6
	Stride time (s)	1.14 (0.11)	1.16 (0.11)	0.344	0.92 [0.82–0.96]	0.09	7.6
	Swing time (s)	0.39 (0.04)	0.39 (0.04)	0.956	0.91 [0.80–0.96]	0.03	8.3
	Stance time (s)	0.77 (0.08)	0.78 (0.08)	0.139	0.92 [0.82–0.97]	0.06	8.0
	Single support time (s)	0.39 (0.04)	0.39 (0.04)	0.861	0.91 [0.81–0.96]	0.03	8.8
	Double support time (s)	0.36 (0.05)	0.37 (0.05)	0.087	0.88 [0.73–0.95]	0.05	12.9
Dual-task	Stride velocity (cm/s)	73.0 (26.0)	75.0 (24.3)	0.449	0.93 [0.85–0.97]	18.03	24.5
	Cadence (step/min)	87.0 (21.0)	88.8 (19.7)	0.343	0.94 [0.87–0.98]	13.48	15.3
	Stride length (cm)	99.8 (21.9)	100.3 (22.7)	0.801	0.95 [0.89–0.98]	13.53	13.5
	Step time (s)	0.74 (0.20)	0.73 (0.19)	0.657	0.93 [0.85–0.97]	0.14	18.7
	Stride time (s)	1.49 (0.44)	1.44 (0.40)	0.968	0.90 [0.78–0.96]	0.37	25.0
	Swing time (s)	0.51 (0.16)	0.49 (0.14)	0.382	0.91 [0.80–0.96]	0.12	24.7
	Stance time (s)	0.96 (0.25)	0.96 (0.27)	0.607	0.92 [0.81–0.96]	0.21	21.8
	Single-support time (s)	0.50 (0.14)	0.49 (0.14)	0.510	0.94 [0.86–0.97]	0.10	19.4
	Double-support time (s)	0.46 (0.13)	0.46 (0.15)	0.737	0.80 [0.53–0.91]	0.17	38.2

SD: standard deviation; CI: confidence interval; ICC: intraclass correlation coefficient; MDC: minimal detectable change.

**Table 3 sensors-22-06392-t003:** Test–retest reliability of spatio-temporal parameters computed by the electronic walkway (reference system) during single and dual-tasking conditions.

Condition	Gait Parameter	Visit T1 Mean (SD)	Visit T2 Mean (SD)	T1 vs. T2 *p*-Value	ICC [95%CI]	MDC	MDC%
Single-task	Stride velocity (cm/s)	97.6 (25.0)	95.7 (24.6)	0.242	0.97 [0.94–0.99]	18.20	24.6
	Cadence (step/min)	105.8 (9.1)	104.6 (9.9)	0.317	0.91 [0.80–0.96]	7.92	7.5
	Stride length (cm)	109.8 (22.8)	109.1 (22.8)	0.455	0.99 [0.97–0.99]	6.93	6.3
	Step time (s)	0.57 (0.05)	0.58 (0.06)	0.291	0.92 [0.81–0.96]	0.05	7.8
	Stride time (s)	1.15 (0.11)	1.16 (0.12)	0.250	0.92 [0.82–0.96]	0.09	7.7
	Swing time (s)	0.40 (0.03)	0.40 (0.03)	0.928	0.88 [0.72–0.95]	0.03	6.9
	Stance time (s)	0.75 (0.09)	0.76 (0.10)	0.206	0.94 [0.87–0.98]	0.06	8.5
	Single support time (s)	0.40 (0.03)	0.39 (0.03)	0.830	0.87 [0.70–0.94]	0.03	7.1
	Double support time (s)	0.36 (0.09)	0.36 (0.09)	0.058	0.98 [0.95–0.99]	0.04	10.3
Dual-task	Stride velocity (cm/s)	74.3 (26.6)	75.7 (25.6)	0.561	0.94 [0.88–0.98]	17.13	22.8
	Cadence (step/min)	87.6 (20.9)	89.1 (20.0)	0.436	0.94 [0.87–0.98]	13.42	15.2
	Stride length (cm)	101.6 (24.2)	101.4 (24.8)	0.989	0.98 [0.95–0.99]	10.51	10.4
	Step time (s)	0.74 (0.21)	0.72 (0.20)	0.932	0.93 [0.83–0.97]	0.16	21.3
	Stride time (s)	1.48 (0.44)	1.44 (0.41)	0.989	0.91 [0.80–0.96]	0.35	23.8
	Swing time (s)	0.49 (0.14)	0.47 (0.12)	0.476	0.94 [0.87–0.97]	0.09	18.9
	Stance time (s)	0.97 (0.29)	0.96 (0.30)	0.732	0.88 [0.73–0.95]	0.29	29.5
	Single-support time (s)	0.48 (0.13)	0.47 (0.11)	0.536	0.87 [0.71–0.94]	0.12	25.5
	Double-support time (s)	0.50 (0.20)	0.49 (0.22)	0.619	0.86 [0.67–0.94]	0.22	44.7

SD: standard deviation; CI: confidence interval; ICC: intraclass correlation coefficient; MDC: minimal detectable change.

**Table 4 sensors-22-06392-t004:** Concurrent validity of pressure-sensing insoles and reference electronic walkway during single-task and dual-task conditions in the mean stride data analysis.

Gait Parameter	Single-Task Condition		Dual-Task Condition	
	Bias [95% LoA]	Corr	Bias [95% LoA]	Corr
Stride velocity (cm/s)	2.50 [−4.17,9.17]	0.99	0.97 [−4.09,6.02]	1.00
Cadence (step/min)	0.27 [−2.70,3.25]	0.99	0.42 [−1.14,1.98]	1.00
Stride length (cm)	2.89 [−4.04,9.82]	0.99	1.45 [−6.67,9.56]	0.99
Step time (s)	−0.01 [−0.04,0.02]	0.96	0.00 [−0.10,0.09]	0.97
Stride time (s)	0.00 [−0.04,0.04]	0.98	−0.02 [−0.02,0.02]	1.00
Swing time (s)	0.00 [−0.06,0.06]	0.61	−0.02 [−0.10,0.06]	0.94
Stance time (s)	−0.02 [−0.10,0.06]	0.92	0.01 [−0.10,0.11]	0.99
Single support time (s)	0.00 [−0.06,0.06]	0.68	−0.02 [−0.14,0.10]	0.90
Double support time (s)	−0.01 [−0.13,0.11]	0.73	0.04 [−0.14,0.22]	0.94

Corr: correlation coefficient; LoA: limits of agreement.

**Table 5 sensors-22-06392-t005:** Concurrent validity of pressure-sensing insoles and reference electronic walkway during single-task and dual-task condition in the stride-to-stride data analysis.

Gait Parameter	Single-Task Condition		Dual-Task Condition	
	Bias [95% LoA]	Corr	Bias [95% LoA]	Corr
Stride velocity (cm/s)	2.04 [−9.11,13.20]	0.97	0.35 [−9.98,10.68]	0.98
Cadence (step/min)	0.16 [−10.43,10.75]	0.87	0.49 [−9.72,10.71]	0.96
Stride length (cm)	2.48 [−9.13,14.10]	0.96	0.37 [−12.63,13.37]	0.95
Step time (s)	−0.01 [−0.10,0.08]	0.79	−0.01 [−0.21,0.19]	0.84
Stride time (s)	0.00 [−0.08,0.08]	0.94	0.00 [−0.11,0.11]	0.99
Swing time (s)	0.00 [−0.08,0.07]	0.61	−0.03 [−0.15,0.10]	0.86
Stance time (s)	−0.01 [−0.12,0.11]	0.84	0.01 [−0.22,0.24]	0.88
Single support time (s)	0.00 [−0.09,0.09]	0.54	−0.03 [−0.19,0.14]	0.73
Double support time (s)	0.01 [−0.13,0.14]	0.69	0.05 [−0.17,0.27]	0.83

Corr: correlation coefficient; LoA: limits of agreement.

## Data Availability

The data presented in this study are available on request from corresponding author.

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
