# Peer review of "Validation of Pressure-Sensing Insoles in Patients with Parkinson’s Disease during Overground Walking in Single and Cognitive Dual-Task Conditions"

_sensors, 2022, doi:10.3390/s22176392_

Round 1

Reviewer 1 Report

In the manuscript entitled “Validation of pressure-sensing insoles in patients with Parkinson’s disease during overground walking in single- and cognitive dual-task conditions.”, the authors propose to examine the validity and reliability of gait parameters computed by pressure-sensing insoles (FeetMe® insoles, France). The results show that spatiotemporal gait parameters were measured simultaneously using the pressure-sensing insoles and the electronic walkway. The concept and demonstrations were also introduced to show its novelty and superiority. Considering this work is of great scientific significance and practical application potential, I would like to recommend the publication of this manuscript in Sensors after the following issues are addressed.

1.     It’s very interesting to see how the pressure-sensing insoles were used, details including processing method, versions (it is different for the old tools?), should presented in the manuscript.

2.     Schematics or physical diagrams should be used to present the pressure-sensing insole, this would make it more like a research article than experiment report.

3.     More abundant structural characterization methods should be used to verify the effects and differences of different pressure-sensing insoles in practical applications, and it is also more intuitive and attractive to readers.

4.     Full names of all proper nouns should be indicated when they first appear in the text.

5.     It will be good if the authors can provide more techniques to collect gait parameters, such as electrochemical sensors. Some up to date literatures are recommend to cite: Engineered Regeneration 2 (2021) 175-181 (https://doi.org/10.1016/j.engreg.2021.12.002)

Reviewer 2 Report

General observations

It is a nice study, well written and with interesting results. Some details need to be reviewed.

Major questions

- Figure 1 and Tables 2, 3, 4 and 5: gait speed is the product of cadence by stride length. However it is not intuitive to use different units (gait speed cm/s = cadence: step/min * Stride length: cm). Also, cadence in step/s can be thought of in Hertz.

- Tables 2, 3, 4 and 5: Why divide the stride length into right and left, if the stride is the sum of the steps of the two legs? Does the same refer to stride time and Double support time? Is there any reason for this? If the reason is because the analysis starts on the right leg and then on the left leg, then isn't the same thing being done at different times? Wouldn't it be a variation in time of the same outcome rather than different outcomes (right leg vs left leg)?

- Table 4 and 5: There was an error typing the title of the two tables. Table 3 is only mean stride data analysis and Table 4 is only stride-to-stride data analysis.

Minor questions

- Line 91: ag (age)

- References: In five references there is not issue and number and not pages (4, 9, 11, 12, 35).
